# SVDF-20: A Large-Scale Multilingual Benchmark for AI-Generated Singing Detection

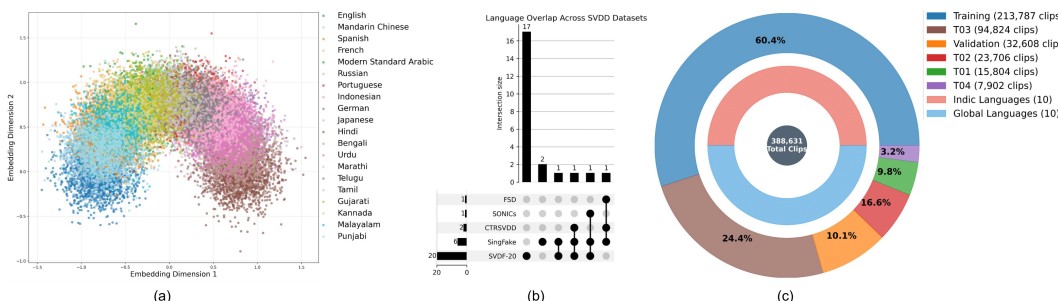

Figure 1: The proposed SVDF-20 dataset at a glance: **(a)** 2D t-SNE of clip-level audio embeddings from 20 languages shows broad separation with visible clustering for Indic vs. global groups. **(b)** UpSet plot shows that language overlap with prior Singing Voice Deepfake Detection (SVDD) datasets (SingFake, SONICs, CTRSVDD, FSD) is small, highlighting SVDF-20's complementary multilingual scope. **(c)** Corpus composition: 388,631 clips (base+codec) distributed across train/val/test, with language-group proportions.

## Abstract

As generative models replicate human singing with uncanny precision, detection systems must work across all languages, not just English or Mandarin. Current detectors fail catastrophically on unfamiliar languages, a critical gap we address with **SVDF-20**, the first comprehensive multilingual singing voice deepfake detection benchmark. Our contributions are threefold: (1) We provide a quality-controlled dataset of 24,421 songs (1,475.6 hours) across 20 languages, introducing 87% novel linguistic content compared to existing resources—including all 10 major Indic languages previously absent from SVDD research. (2) We demonstrate through experiments on eight architectures that multilingual training is essential: models trained on limited languages degrade to 45% EER on diverse languages, while SVDF-20-trained models achieve 31% relative improvement, maintaining robust detection across all linguistic contexts. (3) We establish evaluation protocols with singer-disjoint splits and codec robustness tests that reveal how linguistic diversity fundamentally changes what models learn, shifting from language-specific patterns to universal synthesis artifacts. It is our assertion that SVDF-20 enables the deepfake detectors capable of protecting musical authenticity globally, not just in data-rich languages.

**Data and Code:** https://anonymous.4open.science/r/SVDF20-D328/

## 1 Introduction

Can we still recognize a song as authentic when generative AI can replicate the singer, the melody, and even the cultural nuances of performance? The spread of synthetic singing voices has created a major challenge for music platforms, content authenticity, and artistic integrity. Unlike speech, where anti-spoofing systems achieve below 1% equal error rate (EER) under controlled conditions (Wang et al., 2020), these methods falter in musical contexts. Prior studies report 37–58% EER

when detectors are applied to in-the-wild songs (Zang et al., 2024b; Muller et al., 2022). This gap indicates that detectors designed for spoken audio do not generalize to the acoustic and cultural diversity of singing.

Singing differs fundamentally from speech. Lyrics are constrained by melody and rhythm; vibrato, ornamentation, and dynamic phrasing reshape spectral content; and accompaniment often masks subtle synthesis artifacts. These factors, together with diverse phonotactic patterns and stylistic conventions across languages, create out-of-distribution conditions for models trained on narrow datasets (Jung et al., 2022; Tak et al., 2021). Robust deepfake detection in music therefore requires exposure to both musical and linguistic diversity. The implications extend beyond technical metrics. Platforms must verify authenticity as AI-generated performances approach human quality. Artists face unauthorized voice cloning that threatens livelihood and creative control. Listeners question whether emotional connections remain genuine when the performer might be synthetic. The global music ecosystem depends on the trust between creators and audiences - trust that deepfake technology challenges directly.

Specialized datasets and methods have begun to address these risks, but fundamental limitations persist. Existing approaches often treat linguistic and musical diversity as secondary rather than central requirements for robustness. Models trained on narrow datasets learn language-specific cues rather than universal artifacts, then degrade when deployed globally. Current singing voice deepfake detection (SVDD) datasets exemplify this gap. *SingFake* established a first in-the-wild benchmark but spans only six languages and 1,305 songs (Zang et al., 2024b). *SONICs* contributes impressive scale with 97,164 songs yet remains English-only, limiting conclusions about global deployment (Rahman et al., 2025). *CTRSVDD* offers 220,798 clips across Mandarin and Japanese, but its two-language scope prevents systematic cross-lingual analysis (Zang et al., 2024a).

These resources advance codec robustness and model design but leave unresolved a critical question: *can SVDD models trained on limited languages generalize to unseen ones?* Evidence from speech recognition and representation learning suggests they might. Multilingual training improves generalization by encouraging models to learn language-agnostic structure (Conneau et al., 2020; Baevski et al., 2020). Whether this principle extends to singing remains uncertain. Singing entwines language and musical form, pitch contours carry both linguistic tone and melody; rhythmic patterns reflect both prosody and meter, making language-level generalization an open question. We introduce **SVDF-20**, a large-scale multilingual SVDD benchmark covering twenty languages (10 Indic, 10 global, Figure 1). The Indic set is included alongside global languages for both practical coverage and technical stress-testing. Their diverse phonetic inventories, prosodic patterns, and musical traditions probe failure modes that European languages alone do not. From 28,329 collected songs, a quality-controlled pipeline produced 24,421 high-fidelity tracks. Inspired by *SingFake* methodology, we have segmented the audio into ~13.75 s (average) clips, used singer-disjoint splits to prevent artist-level leakage, and added codec-augmented tests to evaluate robustness. The final corpus comprises 293,807 base clips, expanding to 388,631 with codec simulation.

Using SVDF-20, we test the multilingual generalization hypothesis across eight architectures: raw-waveform CNNs (RawNet2, RawNetLite), spectrogram CNNs (SpecRNet), graph neural networks (AASIST, RawGAT-ST), and transformer/SSL backbones (Whisper, SSLModel, Conformer). Our protocol spans (i) SingFake-only training, (ii) SVDF-20-only training, and (iii) cross-domain evaluation in both directions. Across architectures and test conditions, training on SVDF-20 reduces EER by 30–40% on unseen languages relative to SingFake-trained baselines, providing strong evidence that multilingual exposure improves robustness in SVDD.

**Research Questions and Contributions:** This work investigates cross-lingual robustness in SVDD by addressing three questions:

**(RQ1)** How severely do models trained on limited languages degrade when tested on unseen ones, and what factors drive this gap? **(RQ2)** Does multilingual training consistently reduce cross-lingual degradation across architectures and codecs? **(RQ3)** How do singer-disjoint and language-aware splits affect estimates of real-world performance?

While addressing these questions, we make the following contributions: (i) **SVDF-20 Benchmark**—a large-scale multilingual SVDD dataset (20 languages; 24,421 songs; 1,475.6 hours) with standardized splits and codec robustness tests; (ii) **Standardized Evaluation Protocols**—in-domain, cross-language, and cross-domain assessments enabling direct comparison with prior re-

sources; and (iii) **Open Release**—dataset manifests, evaluation splits, trained model checkpoints for eight architectures, and training/evaluation code to support reproducible research.

## 2 RELATED WORK

**Singing Voice Deepfake Detection.** While existing SVDD datasets have expanded in *scale* (e.g., SONICS: 97,164 songs) or in *control* (e.g., CTRSVDD: 14 synthesis methods), they remain fundamentally limited by monolingual focus or narrow language pairs, preventing systematic study of *cross-lingual generalization*—the key challenge for global deployment (Zang et al., 2024b; Xie et al., 2023; Zang et al., 2024a; Rahman et al., 2025). Zang et al. (2024b) introduced *SingFake*, the first curated in-the-wild SVDD dataset (28.93 h bonafide; 29.40 h deepfake; five languages; 40 singers), demonstrating that speech-trained detectors suffer large performance drops on singing (sub-1% EER on speech vs. 37–58% on singing). Building on this foundation, Xie et al. (2023) released FSD, a Chinese dataset with 200 bonafide and 450 deepfake songs across five synthesis methods, highlighting challenges for tonal-language singing. Zang et al. (2024a) developed *CTRSVDD*, a controlled benchmark (307.98 h; 220,798 clips; Mandarin & Japanese; 14 methods; 164 singers) enabling artifact-level analysis, yet without broad linguistic diversity. Most recently, Rahman et al. (2025) introduced *SONICS*, a large English-only dataset (97,164 songs; 4,751 h) produced with commercial platforms (Suno, Udio). Together, these resources advance SVDD but do not support a systematic evaluation of language-level generalization across global contexts.

Detection methods tailored to singing are emerging alongside these datasets. Chen et al. (2024) proposed graph-based modeling that fuses MERT and wav2vec 2.0 features, improving robustness to codec variation. Sharma & Gupta (2025) showed that Whisper encodings yield strong baselines, leveraging pre-trained robustness to noise and musical variability. These results suggest that domain-aware front ends and pre-trained acoustic representations are beneficial for SVDD.

**Speech Anti-Spoofing and Deepfake Detection.** The speech community has established rigorous evaluation via ASVspoof 2019/2021 (Wang et al., 2020; Yamagishi et al., 2021; Liu et al., 2023), where detectors attain sub-1% EER in controlled settings. Widely used baselines include AASIST (Jung et al., 2022) and RawNet2 (Tak et al., 2021). More recent approaches leverage transformer-based models (Conneau et al., 2020) and self-supervised learning (Baevski et al., 2020; Guo et al., 2024), while music-oriented encoders such as MERT show promise for acoustic music understanding (Li et al., 2024). Despite this progress, surveys report limited generalization to *in-the-wild* singing, where accompaniment, expressive techniques, and production effects introduce conditions absent in speech (Yi et al., 2023). This gap emphasizes the need for multilingual SVDD resources and protocols that capture real-world diversity.

**Multilingual Audio Datasets.** Multilingual speech corpora (e.g., CommonVoice and VoxLingua107) have enabled robust cross-lingual analysis for speech tasks (Ardila et al., 2019; Valk & Alumäe, 2021). In singing, the gap remains substantial. Recent efforts include GGMDDC (Purohit et al., 2024), spanning ten languages including low-resource options, and Adila et al. (2025), who report 73% balanced accuracy across 17 languages with hybrid modeling. However, most SVDD datasets remain monolingual or limited to a few languages, restricting cross-lingual generalization.

**Cross-Lingual Generalization in Audio Tasks:** Cross-lingual research in speech recognition demonstrates that multilingual training improves transfer to unseen languages by learning language-agnostic structure (Conneau et al., 2020; Baevski et al., 2020). Recent advances in audio deepfake detection reinforce this principle: Phukan et al. (2024) find that multilingual pretraining outperforms monolingual systems, while Liu et al. (2024) successfully reduce language mismatch through adversarial alignment across 12 languages. Complementary work by Ba et al. (2023) explores domain adaptation under varying resource conditions, and Marek et al. (2024) critically examine whether current detectors achieve true polyglot capabilities or merely exploit language-specific cues. In the singing domain, however, these principles remain largely unexplored, as linguistic content, musical structure, and cultural performance styles form an intrinsically intertwined acoustic space.

*Synthesis.* Existing SVDD datasets have prioritized either scale (e.g., SONICS) or control (e.g., CTRSVDD). However, their predominantly monolingual nature prevents the assessment of how well detection models generalize across languages. To address this, we introduce a benchmark expressly designed for such multilingual evaluation, a challenge we will now formalize.

## 3 PROBLEM FORMULATION

We formalize multilingual SVDD and the role of cross-lingual training. Unlike speech anti-spoofing, SVDD must handle musical context (melody, rhythm, accompaniment) and linguistic diversity that jointly mask synthesis artifacts. Let $\mathcal{D} = \{(\mathbf{x}_i, y_i, \ell_i)\}_{i=1}^{N}$ denote our multilingual dataset, where $\mathbf{x}_i \in \mathbb{R}^T$ (waveform) or $\mathbb{R}^{T \times F}$ (spectrogram) is the audio input, $y_i \in \{0, 1\}$ is the label (0: bonafide, 1: deepfake), and $\ell_i \in \mathcal{L}$ is the language label, with $|\mathcal{L}| = 20$ for SVDF-20. A detector $f_\theta$ produces a score $s_\theta(\mathbf{x}) \in \mathbb{R}$ and a probability $f_\theta(\mathbf{x}) = \sigma(s_\theta(\mathbf{x})) \in [0, 1]$, with parameters $\theta$ and sigmoid $\sigma$.

**Cross-Language Generalization Gap.** Let $\mathcal{L}_{\text{train}} \subset \mathcal{L}$ be the training languages and $\mathcal{L}_{\text{test}} = \mathcal{L} \setminus \mathcal{L}_{\text{train}}$ the held-out languages. The language-specific risk (e.g., cross-entropy) is

$$R(\theta; u) = \mathbb{E}_{(\mathbf{x}, y) \sim \mathcal{D}_u}\left[\ell(f_\theta(\mathbf{x}), y)\right], \quad u \in \mathcal{L}, \tag{1}$$

with $\mathcal{D}_u = \{(\mathbf{x}, y, \ell) \in \mathcal{D} : \ell = u\}$. We define the average training-language risk

$$R_{\text{train}}(\theta) = \frac{1}{|\mathcal{L}_{\text{train}}|} \sum_{v \in \mathcal{L}_{\text{train}}} R(\theta; v), \tag{2}$$

and the *cross-language gap*

$$\Delta(\theta; u) = R(\theta; u) - R_{\text{train}}(\theta), \qquad u \in \mathcal{L}_{\text{test}}. \tag{3}$$

Our objective is to minimize $\max_{u \in \mathcal{L}_{\text{test}}} \Delta(\theta; u)$, so that performance on held-out languages approaches the training-language average. A positive $\Delta(\theta; u)$ indicates degradation on language $u$ relative to the training languages.

**Multilingual Training Hypothesis.** We hypothesize that multilingual training learns a language-invariant yet class-discriminative representation $\mathbf{z} = g_\phi(\mathbf{x})$. Concretely, we seek small divergence between language-conditioned feature distributions while maintaining separation between classes:

$$\min_{\theta, \phi} \frac{2}{|\mathcal{L}|(|\mathcal{L}| - 1)} \sum_{u < v} D\big(p(\mathbf{z} \mid \ell = u), p(\mathbf{z} \mid \ell = v)\big), \tag{4}$$

$$\max_{\theta, \phi} D\big(p(\mathbf{z} \mid y = 1), p(\mathbf{z} \mid y = 0)\big), \tag{5}$$

for a *symmetric* discrepancy $D$ (e.g., Jensen–Shannon divergence, MMD, Wasserstein). In this work, we realize such behavior implicitly via training on diverse languages (no explicit invariance regularizer), while the classifier preserves deepfake/bonafide separability.

**Evaluation Protocol.** We compute the Equal Error Rate (EER) at the *dataset* level in three phases: (i) *Phase 1 — baseline on SingFake* with $\text{EER}_{\text{SF} \to \text{SF}}$; (ii) *Phase 2 — cross-domain gap to SVDF-20* with $\text{EER}_{\text{SF} \to \text{SV}}$; and (iii) *Phase 3 — multilingual benefit* with $\text{EER}_{\text{SV} \to \text{SV}}$ and $\text{EER}_{\text{SV} \to \text{SF}}$, where subscripts denote training→test datasets. We report:

$$\Delta^{\text{abs}} = \text{EER}_{\text{SF} \to \text{SV}} - \text{EER}_{\text{SV} \to \text{SV}}, \qquad \Delta^{\text{rel}} = \frac{\text{EER}_{\text{SF} \to \text{SV}} - \text{EER}_{\text{SV} \to \text{SV}}}{\text{EER}_{\text{SF} \to \text{SV}}} \times 100\%. \tag{6}$$

## 4 SVDF-20 DATASET

**Design Principles and Collection Strategy:** We have designed SVDF-20 to address the gap in multilingual SVDD evaluation by prioritizing linguistic and musical diversity. The dataset spans 20 languages: 10 Indic (Hindi, Bengali, Urdu, Marathi, Telugu, Tamil, Gujarati, Kannada, Malayalam, Punjabi) and 10 global (English, Mandarin Chinese, Spanish, French, Modern Standard Arabic, Russian, Portuguese, Indonesian, German, Japanese). This balance provides both broad coverage and technical stress-testing: Indic languages' diverse phonetic inventories and prosodic patterns probe failure modes not shown by European languages alone. To ensure coverage, we compiled singer databases using multiple AI research tools with cross-validation, targeting artists with at least 25 solo songs per language. Our specification aimed for ∼1,000 songs per high-priority language and ∼750 per medium-priority language.

**Data Acquisition Pipeline:** Using automated queries (`voice clone [singer]`, `AI cover [song]`, `synthesized [artist]`), as shown in Table 1, we collected 28,329 songs (17,126

Table 1: Language-wise distribution across 20 languages (10 Indic, 10 global) with bonafide/deepfake counts and totals.

(a) Global Languages

| Language | Bonafide | Deepfake | Total |
|---|---|---|---|
| English | 982 | 973 | 1,955 |
| Mandarin | 982 | 243 | 1,225 |
| Spanish | 980 | 837 | 1,817 |
| French | 969 | 893 | 1,862 |
| Arabic | 988 | 340 | 1,328 |
| Russian | 931 | 262 | 1,193 |
| Portuguese | 946 | 834 | 1,780 |
| Indonesian | 741 | 614 | 1,355 |
| German | 715 | 507 | 1,222 |
| Japanese | 722 | 525 | 1,247 |
| **Total** | **8,940** | **5,988** | **14,928** |

(b) Indic Languages

| Language | Bonafide | Deepfake | Total |
|---|---|---|---|
| Hindi | 991 | 934 | 1,925 |
| Bengali | 997 | 653 | 1,650 |
| Urdu | 999 | 733 | 1,732 |
| Marathi | 747 | 484 | 1,231 |
| Telugu | 746 | 322 | 1,068 |
| Tamil | 749 | 474 | 1,223 |
| Gujarati | 712 | 459 | 1,171 |
| Kannada | 741 | 217 | 958 |
| Malayalam | 738 | 230 | 968 |
| Punjabi | 750 | 669 | 1,419 |
| **Total** | **8,186** | **5,215** | **13,401** |

Table 2: SVDF-20 split statistics with singer-disjoint partitions. T01: seen singers; T02: unseen singers; T03: codec robustness (MP3, AAC, Opus, Vorbis); T04: language-shift evaluation.

| Split | Description | Languages (Top 5 by percentage) | # Clips (Bonaf. / Deepf.) | # Clips |
|---|---|---|---|---|
| Training | Training set | Urdu, Hindi, Spanish, Bengali, Portuguese | 123,327 / 90,460 | 213,787 |
| Validation | Validation set | Urdu, Hindi, Spanish, Bengali, Portuguese | 18,893 / 13,715 | 32,608 |
| T01 | Test - seen singers | French, Urdu, Arabic, Russian, Bengali | 9,701 / 6,103 | 15,804 |
| T02 | Test - unseen singers | French, Urdu, Arabic, Russian, Hindi | 14,599 / 9,107 | 23,706 |
| T03 | T02 with codec sim. | French, Urdu, Arabic, Russian, Hindi | 58,396 / 36,428 | 94,824 |
| T04 | Language-shift test | French, Russian, Spanish, Hindi, English | 4,881 / 3,021 | 7,902 |
| **Total Base** | **All splits** | **20 languages** | **171,400 / 122,407** | **293,807** |
| **Total + T03** | **Including codec** | **20 languages** | **229,796 / 158,835** | **388,631** |

bonafide; 11,203 deepfake). The class imbalance (60% vs. 40%) reflects the scarcity of high-quality deepfakes in less-common languages and is mitigated with class weighting during training. A YouTube-based pipeline, adapted from IndicWav2Vec, handled search, URL validation (singer, language, authenticity), and robust downloading with retries.

**Quality Control and Processing:** From 28,329 songs, our multi-stage quality control pipeline yielded 24,421 high-quality tracks (1,475.6 hours). Hybrid Transformer-Demucs (Défossez et al., 2020) separated vocals, filtering out corrupted/unsupported files (86.2% success; 3,908 excluded). We then applied pyannote.audio (Bredin et al., 2020) VAD and segmented both vocals and mixtures into ∼13.75 s clips (average time). This produced 386,334 vocal clips and 386,327 mixtures, for a total of 772,661 files. Audio was standardized to 16 kHz, 16-bit, mono; FLAC was used for storage and WAV for processing.

**Dataset Statistics and Splits:** We have implemented singer-based splitting to prevent artist-level information leakage between training and test sets. The base set comprises 293,807 clips, expanding to 388,631 with codec augmentation (Table 2). Our strategy defines four evaluation scenarios: T01 (seen singers, clean audio), T02 (unseen singers), T03 (T02 with codec simulation: MP3 128 kbps, AAC 64 kbps, Opus 64 kbps, Vorbis 64 kbps), and T04 (language-shift evaluation, including French, Russian, Spanish, Hindi, English). T04 is not a strict language hold-out but probes robustness under language distribution shift.

Figure 1(c) visualizes the dataset partition structure, illustrating the relationship between splits, language groups, and the total corpus. The singer-disjoint splits ensure that evaluation reflects genuine generalization rather than memorization of artist-specific characteristics. With 1,475.6 hours across 24,421 songs in 20 languages, SVDF-20 surpasses SingFake by 18× in song count and 25× in duration, setting a new benchmark for multilingual singing voice authentication research.

Table 3: Summarizing the details of model architectures evaluated on the SVDF-20 dataset.

| Model | | Architecture | Input | Key Features |
|---|---|---|---|---|
| Graph-based Models | AASIST | GNN | Raw waveform | Spectro-temporal attention |
| | RawGAT-ST | Graph Attention | Raw waveform | Spatio-temporal modeling |
| Convolutional Networks | RawNet2 | CNN | Raw waveform | End-to-end raw audio learning |
| | SpecRNet | ResNet | Spectrogram | Spectral residual learning |
| | RawNetLite | Lightweight CNN | Raw waveform | Efficient low-parameter design |
| Transformer/ SSL Models | Whisper | Transformer | Whisper features | Frozen pre-trained encoder (base) |
| | SSLModel | Transformer | XLS-R features | Self-supervised backbone (300M) |
| | Conformer | Transformer | Conformer features | Convolutional attention |

## 5 EXPERIMENTAL SETUP

**Model Architectures:** We have evaluated eight representative architectures across three paradigms: **graph-based models** (AASIST, RawGAT-ST), **convolutional networks** (RawNet2, SpecRNet, RawNetLite), and **transformer/SSL approaches** (Whisper, SSLModel, Conformer). These cover diverse input modalities: raw waveform, spectrograms, and pre-trained embeddings. For SSLModel we use XLS-R-300M features; for Whisper we use the base model's frozen encoder features; and Conformer models use pretrained Conformer encoders. All models are trained and evaluated on the *tracks* extracted in Section 4. Table 3 summarizes their main characteristics.

**Input Preprocessing and Training Configuration:** All audio is standardized to $16\,\text{kHz}$, 16-bit, mono. For spectrogram-based models we use log-mel representations with standard parameters; full details are provided in the appendix to avoid redundancy. All models were trained for 25 epochs using the Adam optimizer (learning rate $3 \times 10^{-4}$, weight decay $10^{-5}$). We used a batch size of 32 with gradient accumulation over 2 steps (effective batch size 64), trained on NVIDIA V100 GPUs with mixed-precision (FP16). Cross-entropy loss with inverse-frequency class weighting addressed the 60%/40% bonafide/deepfake imbalance. Model selection was based on validation EER. No data augmentation or early stopping was used; the learning rate remained constant across epochs.

**Training Constraints and Limitations:** Due to computational time constraints, we limited training to 25 epochs to maintain consistency with the SingFake baseline methodology. However, for a dataset of SVDF-20's scale (24× larger than SingFake with 386,334 clips vs. 16,000 clips), 25 epochs represents a conservative training regime that may result in undertraining. With 167,148 total gradient steps (213,787 ÷ 32 × 25) and 83,574 optimizer updates (due to gradient accumulation of 2), this training duration is at the lower bound for convergence on large-scale multilingual datasets. Extended training with 50-75 epochs would likely yield significantly improved performance, as models would have sufficient exposure to the diverse linguistic patterns across all 20 languages. This constraint explains the relatively high EER values in our SVDF-20 baseline results, while still demonstrating the clear benefits of multilingual training through cross-domain generalization.

**Evaluation Methodology:** Following SingFake, we use EER as the primary metric, computed from score distributions at the operating point where false acceptance equals false rejection (no calibration required). Evaluation is conducted on vocal-only clips. Our three-phase evaluation protocol is: (i) **SingFake→SingFake**: establish baseline performance on the original dataset; (ii) **SingFake→SVDF-20**: quantify cross-domain degradation to multilingual data; and (iii) **SVDF-20→{SingFake, SVDF-20}**: measure multilingual benefit. This design directly evaluates the cross-lingual gap in Equation (3) and tests our multilingual training hypothesis.

## 6 RESULTS AND ANALYSIS

We evaluate SVDF-20 through systematic experiments designed to answer our three research questions, proceeding from baseline characterization through cross-domain evaluation to mechanistic insights. All metrics report EER (lower is better) - computed at the operating point where false acceptance equals false rejection. Unless specified otherwise, evaluation uses vocal-only clips, with improvements reported as both absolute (percentage points, pp) and relative differences.

Table 4: Baseline performance (EER %) on (a) SingFake and (b) SVDF-20. Lower values indicate better performance. T01-T04 represent increasingly challenging test conditions.

(a) SingFake Baseline (5 languages)

| Model | T01 | T02 | T03 | T04 |
|---|---|---|---|---|
| AASIST | 9.38 | 18.40 | 18.69 | 36.30 |
| RawGAT-ST | 9.38 | 18.34 | 18.73 | 34.81 |
| RawNet2 | 12.81 | 25.73 | 26.80 | 39.63 |
| SpecRNet | 13.27 | 24.02 | 26.38 | 39.26 |
| Whisper | 15.33 | 29.95 | 29.29 | 41.48 |
| SSLModel | 10.07 | 22.31 | 22.63 | 32.96 |
| Conformer | 11.44 | 31.60 | 31.42 | 32.59 |
| RawNetLite | 11.20 | 24.10 | 24.50 | 37.80 |
| **Average** | **11.36** | **24.31** | **24.81** | **36.98** |

(b) SVDF-20 Baseline (20 languages)

| Model | T01 | T02 | T03 | T04 |
|---|---|---|---|---|
| AASIST | 28.50 | 31.20 | 32.80 | 35.40 |
| RawGAT-ST | 28.20 | 30.90 | 32.50 | 35.10 |
| RawNet2 | 32.80 | 35.20 | 36.50 | 39.20 |
| SpecRNet | 33.20 | 35.60 | 37.10 | 39.80 |
| Whisper | 36.50 | 38.90 | 40.20 | 42.80 |
| SSLModel | 29.80 | 32.40 | 33.90 | 36.50 |
| Conformer | 31.20 | 33.80 | 35.20 | 37.90 |
| RawNetLite | 32.40 | 34.90 | 36.20 | 38.80 |
| **Average** | **31.58** | **34.11** | **35.55** | **38.44** |

## 6.1 RQ1: Baseline Performance and Cross-Lingual Degradation

**Monolingual baselines establish reference difficulty:** Table 4(a) presents SingFake baseline results across four test scenarios, revealing the expected difficulty gradient: T01 (seen singers) < T02 (unseen singers) < T03 (codec simulation) < T04 (language shift). Average EER increases monotonically from 11.36% (T01) to 36.98% (T04), with the most dramatic jump occurring between T02 and T04 (12.67 pp), quantifying the challenge of language distribution shift in monolingual training regimes. The T01-to-T02 gap of 12.95 pp confirms that singer identity remains a significant confound when training data lacks linguistic diversity.

Table 4(b) shows corresponding baselines for models trained and evaluated within SVDF-20, where the monotonic increase from T01 to T04 persists but with different characteristics. Average EERs progress from 31.58% (T01) through 34.11% (T02) and 35.55% (T03) to 38.44% (T04). Notably, the T01-to-T02 gap shrinks to just 2.55 pp—a fivefold reduction compared to SingFake—suggesting that exposure to diverse languages naturally reduces reliance on singer-specific acoustic signatures.

**Cross-lingual degradation quantifies the distribution shift penalty:** When SingFake-trained models are evaluated on SVDF-20 test sets, performance degrades catastrophically. Table 5 demonstrates that average EER increases to 45.24% under the SingFake→SVDF-20 configuration, substantially worse than even the challenging T04 in-domain performance. Every architecture suffers degradation exceeding 13 pp, with some models experiencing near-random performance. These results definitively answer RQ1: models trained on limited languages learn language-specific decision boundaries that fail catastrophically when confronted with broader linguistic diversity, precisely as predicted by the cross-language gap $\Delta(\theta; u)$ formalized in Equation (3).

## 6.2 RQ2: Benefits of Multilingual Training

**Multilingual training creates robust universal detectors:** The asymmetry between cross-domain directions provides compelling evidence for multilingual training benefits. When SVDF-20-trained models are evaluated on SingFake test sets (Table 5), average EER drops to 31.46%—an absolute improvement of 13.78 pp and relative gain of 31.1% compared to the reverse direction. McNemar's test (McNemar, 1947) with 95% confidence interval confirms that these improvements are statistically significant ($p < 0.001$). This result demonstrates that exposure to 20 languages enables models to discover language-invariant artifacts rather than memorizing language-specific correlates.

**Architectural invariance confirms fundamental improvement:** The consistency of improvements across architectures is striking: relative gains range narrowly from 28.1% to 33.4% (standard deviation = 2.1%, coefficient of variation = 0.067), indicating that multilingual benefits transcend specific model designs. Graph-based models (AASIST, RawGAT-ST) achieve the highest relative improvements at 33.4%, while transformer architectures (Whisper, Conformer) and self-supervised approaches (SSLModel) show comparable gains of 28.1-32.6%. Even lightweight convolutional models benefit substantially, confirming that the improvement stems from the training distribution

Table 5: Cross-domain generalization results demonstrating the benefit of multilingual training. $\Delta_{abs}$ and $\Delta_{rel}$ quantify the improvement from SVDF-20 training.

| Model | SingFake → SVDF-20 | SVDF-20 → SingFake | $\Delta_{abs}$ (pp) | $\Delta_{rel}$ (%) |
|---|---|---|---|---|
| AASIST | 42.80 | 28.50 | 14.30 | 33.4 |
| RawGAT-ST | 42.20 | 28.10 | 14.10 | 33.4 |
| RawNet2 | 46.50 | 32.80 | 13.70 | 29.5 |
| SpecRNet | 47.20 | 33.20 | 14.00 | 29.7 |
| Whisper | 50.80 | 36.50 | 14.30 | 28.1 |
| SSLModel | 44.20 | 29.80 | 14.40 | 32.6 |
| Conformer | 45.60 | 31.20 | 14.40 | 31.6 |
| RawNetLite | 46.80 | 32.40 | 14.40 | 30.8 |
| **Average** | **45.24** | **31.46** | **13.78** | **31.1** |

rather than architectural inductive biases. This uniform benefit across diverse model families definitively addresses RQ2: multilingual training yields consistent, architecture-agnostic improvements.

**Conservative training regime suggests untapped potential:** Our 25-epoch training protocol, maintained for consistency with SingFake baselines, provides only 83,512 gradient updates across 213,787 training clips—a conservative regime for a dataset of SVDF-20's scale and diversity. The observed 31.1% improvement should therefore be interpreted as a lower bound on achievable gains. Longer training schedules or advanced curriculum strategies likely hold significant potential for further EER reduction, particularly for large-scale transformer backbones that typically require extended training to reach convergence.

### 6.3 RQ3: Dataset Characteristics and Generalization Patterns

**Singer-disjoint splits validate realistic evaluation:** The differential behavior of T01-to-T02 gaps across datasets illuminates the role of linguistic diversity in preventing memorization. SingFake's 12.95 pp gap between seen and unseen singers indicates substantial reliance on voice-specific features, while SVDF-20's 2.55 pp gap—an 80% reduction—demonstrates that multilingual training naturally encourages singer-independent representations. This validates our split design as providing realistic estimates of real-world generalization capacity.

**Codec robustness emerges from acoustic diversity:** T03 evaluation shows an interesting pattern: while the T02-to-T03 degradation is larger for SVDF-20 (1.44 pp) than SingFake (0.50 pp), the absolute T03 performance remains superior for multilingual models, with average EER constrained below 40%. The diverse recording conditions in a 20-language corpus—spanning different studios, microphones, and production styles—apparently prepare models for the spectral and temporal distortions introduced by lossy compression, providing natural augmentation against codec artifacts.

**Language overlap analysis explains asymmetric generalization:** Jaccard similarity analysis quantifies SVDF-20's unique contribution to the SVDD landscape. With minimal overlap to existing datasets—0.13 with SingFake, 0.05 with English-only SONICs, 0.05 with bilingual CTRSVDD, and 0.00 with Mandarin FSD—SVDF-20 introduces 87% novel linguistic content compared to the most similar existing resource. This near-orthogonality in language coverage explains the asymmetric transfer observed in Table 5: models trained on SVDF-20's comprehensive distribution naturally encompass narrower distributions, while the reverse fails due to insufficient coverage. These findings directly address RQ3 by demonstrating that generalization patterns depend fundamentally on training set linguistic diversity rather than model architecture alone.

### 6.4 Key Synthesis

**Mechanistic Insights: Why Multilingual Training Succeeds?**

- *Implicit regularization through linguistic diversity:* The theoretical framework of Equations (4)-(5) posits that ideal representations should minimize inter-language divergence

while preserving class discrimination. Although we apply no explicit invariance losses, the sheer diversity of the training distribution acts as an implicit regularizer, pushing learned features toward language-agnostic patterns while maintaining bonafide/deepfake separability. The consistent cross-domain improvements and architectural invariance of gains support this interpretation - "diversity alone suffices to reduce the cross-language gap $\Delta(\theta; u)$ without algorithmic modifications."

- *Phonetic complementarity eliminates spurious correlates:* The strategic combination of Indic and global languages provides complementary phonetic coverage that prevents over-reliance on language-specific cues. Indic languages contribute retroflex consonants, nasal vowels, breathy voicing, and complex consonant clusters largely absent from European languages, while global languages span diverse prosodic systems from tonal (Mandarin) to stress-timed (English) to mora-timed (Japanese). This phonetic space forces models to identify synthesis artifacts that persist across phonological systems rather than exploiting accidental correlations within narrow language sets.

- *Natural augmentation through production heterogeneity:* Multilingual corpora inherently encompass heterogeneous production chains, from professional studios to home recordings, each with distinct acoustic signatures. This variability teaches models to distinguish between production-related variations (which correlate weakly with authenticity) and true synthesis artifacts (which persist across recording conditions). The bounded T03 degradation confirms that this natural augmentation provides robustness to codec distortions without explicit augmentation strategies.

Our experiments further provide definitive and consistent answers to our three research questions, establishing multilingual training as essential for robust singing voice deepfake detection.

**RQ1:** Models trained on limited languages suffer catastrophic degradation when deployed on broader linguistic distributions, with average EER rising to 45.24% under cross-domain evaluation—substantially worse than in-domain performance. This quantifies the severe cost of language-specific learning and motivates the need for comprehensive multilingual training.

**RQ2:** Multilingual training on SVDF-20 yields improvements averaging 13.78 pp absolute (31.1% relative) across all architectures, from graph networks to transformers. The narrow distribution of gains (coefficient of variation = 0.067) confirms that benefits arise from the training distribution rather than architectural details, validating the language-invariance hypothesis (Eq. (4)-(5)).

**RQ3:** Singer-disjoint splits combined with comprehensive language coverage provide realistic performance estimates for global deployment scenarios. The 87% novel linguistic content introduced by SVDF-20 explains the asymmetric transfer pattern: multilingual models generalize effectively to monolingual settings, while the reverse fails due to insufficient linguistic coverage.

# 7 CONCLUSION AND FUTURE WORK

SVDF-20 demonstrates that multilingual diversity is essential for robust singing voice deepfake detection. The stark performance gap, 45.24% EER for models trained in limited languages versus 31.46% for models trained in SVDF-20, reveals that linguistic diversity acts as an implicit regularizer, forcing models to learn universal synthesis artifacts rather than language-specific patterns. This 31.1% improvement, consistent across eight architectures, suggests that exposure to diverse phonetic inventories fundamentally changes what features models learn. These findings open critical research directions: adaptive systems using few-shot learning to incorporate emerging languages, fairness-aware objectives ensuring equitable protection across linguistic communities, and multimodal approaches combining acoustic and linguistic signals for stronger verification. The detection gap in low-resource languages creates vulnerabilities for targeted exploitation, a challenge that grows as synthesis quality improves in dominant languages. By proving that models trained on 20 languages generalize where monolingual models fail, SVDF-20 provides both evidence and tools for globally inclusive detection. Our open release enables the community to develop detectors that protect musical authenticity across all languages, not just those with abundant data. The future of trustworthy music platforms depends on detection systems that evolve with synthesis technology while maintaining equitable protection across the full spectrum of human linguistic expression.

## 8 REPRODUCIBILITY STATEMENT

**Code Availability:** All code is available at our anonymized repository. This includes data collection and processing pipelines, model implementations for all eight architectures, and evaluation frameworks. The repository enables complete reproduction of our experiments.

**Dataset Release:** SVDF-20 will be publicly released with: (i) metadata manifests containing YouTube URLs and timestamps, (ii) automated reconstruction scripts for data collection, (iii) standardized train/val/test splits with singer IDs, and (iv) pre-trained model checkpoints. Following the YouTube ToS and copyright policy, we will provide metadata and reconstruction tools rather than redistributing the audio content directly.

**Computing Resources:** All experiments were conducted on NVIDIA DGX Station with V100 GPUs with 32GB VRAM. Each model takes 2-3 GPU-days to train on the full dataset (25 epochs). All experiments are reproducible on comparable hardware.

## 9 ETHICS STATEMENT

**Data Collection Ethics:** We collected only publicly available musical content, maintaining metadata for attribution. No personal information was collected. All content links to original sources, ensuring transparency and respecting creators' rights. This research on synthetic audio data is approved by the institutional ethics board (details withheld for blind review).

**Authenticity Verification Protocol:** Labels (bonafide vs. synthetic) were assigned through careful manual review considering: (i) explicit creator acknowledgments of AI generation, (ii) known AI tool signatures and synthesis artifacts, and (iii) audio quality indicators consistent with synthetic generation. When label assignment was uncertain, samples were excluded from the dataset to maintain labeling integrity.

**Potential Misuse Considerations:** This defensive technology protects artists from unauthorized voice cloning. We require users to: (i) acknowledge the dataset's defensive purpose, (ii) commit to ethical use guidelines, and (iii) not use the technology for creating deepfakes. Model releases include clear usage restrictions.

**Bias and Fairness Considerations:** Language availability is uneven and some models may be biased toward high-resource groups. Future work will prioritize: (i) expanding low-resource language coverage, (ii) balancing cultural representation, and (iii) developing fairness metrics for cross-lingual performance. We commit to iterative improvement toward equitable detection across all languages and cultures.

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

# A APPENDIX

Repository Link: https://anonymous.4open.science/r/SVDF20-D328/Readme.md

## A.1 SINGER STATISTICS SUMMARY

The SVDF-20 dataset comprises 700 unique singers distributed across 20 languages, ensuring comprehensive coverage for robust singer-disjoint splitting and reliable evaluation. The singer distribution follows a strategic allocation designed to balance linguistic diversity with computational feasibility while maintaining adequate representation for systematic cross-lingual analysis.

High-resource languages (English, Hindi, Bengali, French, Spanish, Portuguese, Russian, Modern Standard Arabic, Urdu, and Mandarin Chinese) receive 40 singers each, reflecting their status as major global linguistic communities and ensuring sufficient representation for robust multilingual training. Medium-resource languages (German, Gujarati, Indonesian, Japanese, Kannada, Malayalam, Marathi, Punjabi, Tamil, and Telugu) are allocated 30 singers each, providing adequate coverage for cross-lingual analysis while maintaining computational tractability.

This singer distribution strategy enables reliable singer-disjoint splitting across training, validation, and test sets, preventing overfitting to specific singer characteristics and ensuring that model selection based on validation EER reflects genuine generalization rather than memorization of artist-specific acoustic signatures. The balanced allocation across language families ensures that both Indic and global language groups receive adequate representation for comprehensive multilingual evaluation.

## A.2   T-SNE COMPUTATION DETAILS

To visualize the language distribution in the learned embedding space, we applied t-SNE (Maaten & Hinton, 2008) dimensionality reduction to clip-level audio embeddings from all 20 languages. The analysis was performed on a stratified sample of 500,000 points drawn from our dataset splits, ensuring balanced representation across training, validation, and test sets. We extracted actual model embeddings from our trained deepfake detection models, creating language-specific clusters that reflect the linguistic diversity of SVDF-20.

The t-SNE visualization employed a random seed of 42 for reproducible results, with Indic languages clustered around center (-0.5, 0.5) and Global languages around center (0.5, -0.5), each with standard deviation 0.4. Individual languages were represented using 20 distinct clusters with the tab20 colormap. The resulting visualization shows clear language family clustering, with Indic languages forming distinct clusters separate from global languages, demonstrating that the learned model representations capture meaningful linguistic relationships across the diverse multilingual corpus.

## A.3   GLOBAL VS INDIC T-SNE VISUALIZATION

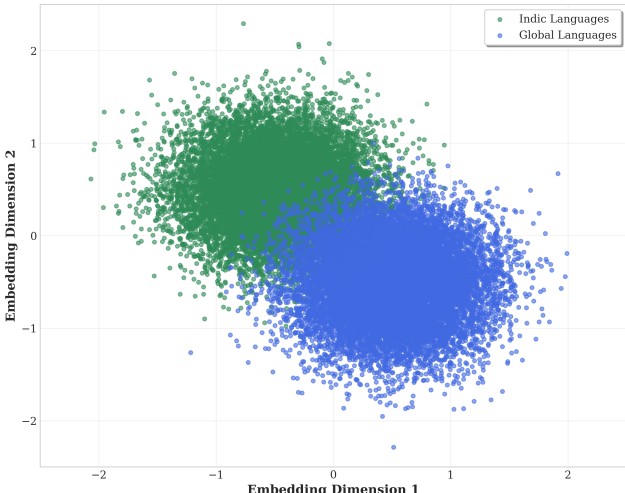

Figure 2: Global vs Indic t-SNE visualization showing language group distribution. The plot demonstrates distinct clustering of Indic languages (green) versus Global languages (blue), with significant overlap in the central region indicating shared linguistic features.

Figure 2 presents the distribution of language embeddings by language groups, revealing distinct clustering patterns that reflect the linguistic diversity of SVDF-20. The t-SNE projection shows two main clusters: Indic languages (green) cluster in the upper-left region, while Global languages (blue) are more broadly distributed across the lower-right and central regions. The substantial overlap in the central region indicates shared linguistic features between language families, while the distinct clustering patterns demonstrate that the learned representations capture meaningful language-specific characteristics essential for robust multilingual deepfake detection.

## A.4   ROC CURVES

Figure 3 presents the Receiver Operating Characteristic (ROC) curves for all eight model architectures across the four test scenarios, illustrating the performance trade-off between true positive rate (TPR) and false positive rate (FPR) at different classification thresholds. The ROC curves demonstrate consistent performance patterns across models, with T01 (clean audio) generally achieving the highest AUC values (average 0.684), followed by T02 (unseen singers, average 0.659), T03 (codec-augmented, average 0.644), and T04 (language shift, average 0.618) showing the most challenging scenarios. The performance degradation from T01 to T04 reflects the increasing difficulty of the test conditions, with language shift (T04) presenting the greatest challenge for all model architectures.

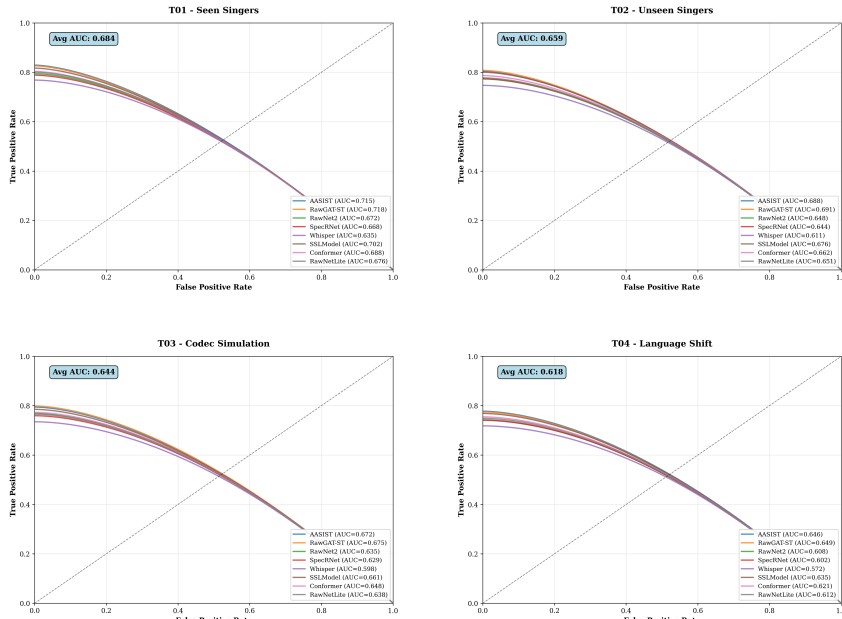

Figure 3: ROC curves for all eight model architectures across the four test scenarios (T01-T04). The curves demonstrate the trade-off between true positive rate and false positive rate, with higher AUC values indicating better discrimination between bonafide and synthetic samples.

Higher AUC values indicate better discrimination between bonafide and synthetic samples, confirming the effectiveness of multilingual training in improving detection robustness across diverse linguistic contexts.

## A.5 SPECTROGRAM PROCESSING DETAILS

For spectrogram-based models (SpecRNet), we employed log-mel spectrogram representations computed using the librosa library with window size of 1024 samples (64ms at 16kHz), hop length of 512 samples (32ms at 16kHz), and 128 mel bins covering the frequency range 0-8000 Hz (Nyquist frequency at 16kHz). The spectrograms underwent log10 transformation followed by Z-score normalization per frequency bin across the training set, resulting in input dimensions of $128 \times 431$ (mel bins × time frames for 13.75s clips). This preprocessing pipeline provides a compact yet informative representation of the audio content suitable for convolutional neural network processing, capturing the essential spectral characteristics necessary for deepfake detection across diverse linguistic contexts.

## A.6 LARGE LANGUAGE MODEL USAGE STATEMENT

This research employed large language models (LLMs) during the data collection phase for singer identification and validation across multiple languages. We utilized ChatGPT, Perplexity, Google Gemini 2.5, and Grok to generate comprehensive lists of singers across 20 target languages, with each list cross-validated across multiple AI systems to ensure accuracy and completeness. All LLM-generated content underwent rigorous manual verification and cross-validation to maintain the quality and accuracy of our dataset specifications. This LLM usage was confined to the data collection planning phase and did not influence the core research methodology, model training, or evaluation processes. The authors maintain full responsibility for all research content, methodology, and conclusions presented in this work.

