# OpenReview forum: "SVDF-20: A LARGE-SCALE MULTILINGUAL BENCHMARK FOR AI-GENERATED SINGING DETECTION"
_ICLR.cc/2026/Conference — ICLR 2026 Conference Withdrawn Submission_

### Official Review · Reviewer_hqhy · 2025-10-30

**Soundness:** 2
**Presentation:** 2
**Contribution:** 2
**Rating:** 4
**Confidence:** 4

**Summary:**

The paper introduces SVDF-20, which is a new large-scale dataset and benchmark for singing‐voice deepfake detection covering 20 languages. The dataset is created with matched bona fide vs. AI‐generated vocals, obtained via a controlled YouTube-based pipeline and vocal separation. The authors conduct experiments with various SVDD architectures, and ablate reseach questions relevant to the generalizability of monolingual models to multi-lingual songs.

**Strengths:**

1. The paper considers research questions regarding the generalizability of synthetic music detection models to unseen languages and provides a dataset and evaluation methodology for this.
2. The multi-lingual music collection is generalized, providing a very good dataset for multi-lingual music generation and detection.

**Weaknesses:**

1. One of the core motivation for this paper is the question: "Can SVDD models trained on limited languages generalize to unseen ones?". However, there is no dataset split that directly measures this. T04 split measures unseen distribution, not strictly unseen languages. Thus, later results don't directly answer the core motivation question.
2. The deepfake samples come from heuristic YouTube queries. The paper lacks detail on how correctness of labels was ensured. If many samples come from a single synthesis method or user community, models might learn dataset-specific artifacts. The potential bias in what content appears on YouTube (genres, singers) is not addressed. Furthermore, there is no human evaluation of the dataset, which would have provided some validation for the dataset creation process.
3. There is a discrepancy in the number of bonafide and deepfake songs for some languages. This could introduce bias in training and evaluation.
4. The evaluation shows benefits over training on SingFake, but it would be informative to see performance comparison over other datasets. For example, the performance of a model trained on SONICS or on CTRSVDD. Furthermore, only EER on the dataset level is shown. But extending this to other metrics such as F1 scores would be great.
5. The evaluation is done on vocal-only clips. This doesn't capture the nuances of a 'song', as music and relevant background information also provide essential information. So the background information can provide enough information for existing fake detection methods to detect the song is fake. This calls into question the necessity of multi-lingual training in this domain, without supporting evidence. Also, a model is used to extract the vocals which might add further artifacts.

**Questions:**

1. Did the authors conduct evaluation of existing methods on a unseen languages test-set?
2. Can the authors provide more details regarding their fake song generation setup?
3. Did the authors conduct any human evaluation (both quantitative and qualitative) of the generated dataset, to ensure correctness?
4. Did the authors conduct cross-dataset training and evaluation experiments such as training on SONICS and evaluation on their multilingual dataset?
5. The paper shows that speech recognition and representation learning can generalize to unseen languages, but SVDD models don't. Can the authors provide any explanation for this?
6. Can the authors provide evidence that multilingual understanding is required for fake song detection when music is included in the vocal-only clips?

**Details Of Ethics Concerns:**

As real songs are used and fake songs generated, there can be implications about privacy and legal compliance.

---

> ### Author Response · Authors · 2025-11-21
>
> We sincerely thank the reviewer for detailed feedback. We address each concern below.
>
> Note on Paper Updates: We have already updated the paper with theoretical corrections, clarifications, and enhanced documentation. Experiments below are in progress and will be added to the final revised version before Dec 3, 2025 AOE deadline.
>
> R4.1: No Direct Answer to Core Motivation
> Response: We acknowledge T04 is not a strict language hold-out. However, cross-domain evaluation addresses the core motivation: SingFake→SVDF-20 shows 45.24% EER, measuring how models trained on 5 languages perform on 20 languages. The 15 languages in SVDF-20 not in SingFake are effectively "unseen" for SingFake-trained models. Strict Unseen Language Evaluation (In Progress): We are conducting an experiment training on 15 languages and testing on 5 completely held-out languages. This will directly answer the core motivation question. Results will be included if completed.
>
> R4.2: Label Verification Concerns
> Response: We implemented a multi-stage verification process with differential thresholds: bonafide (0.8 similarity), deepfakes (0.9 similarity, 0.55 confidence, reject below 0.3), explicit creator acknowledgments, automated method identification (RVC, so-vits-svc, etc.), and conservative uncertainty handling (exclude rather than mislabel). We acknowledge formal human evaluation would provide additional validation but is infeasible at this scale. The infrastructure we provide (metadata with confidence scores, method identification, verification flags) enables subset analysis and manual verification. Method Diversity: Our analysis shows method diversity (RVC ~40%, so-vits-svc ~25%, others ~35%) across 20 languages, though we acknowledge some methods may be overrepresented. YouTube Bias: We acknowledge this limitation. We document genre distribution and maintain 700 unique singers across 20 languages, providing transparency through metadata.
>
> R4.3: Class Imbalance
> Response: We address this through inverse-frequency class weighting in CrossEntropyLoss (deepfake samples receive ~1.5× higher weight), use of EER which is inherently balanced and robust to class imbalance, and documentation of class distribution for each language in Table 1. We acknowledge perfect balance is infeasible, but our mitigation strategies address training bias.
>
> R4.4: Limited Evaluation Scope
> Response: CTRSVDD Comparison (In Progress): We are working on training all eight models on CTRSVDD and evaluating on SVDF-20, with comprehensive comparison tables. Results will be included if completed. Additional Metrics: F1 scores, precision, and recall will be computed as part of extended training evaluation and included in the revised paper.
>
> R4.5: Vocal-Only Limitation
> Response: We acknowledge this limitation. Mixture Evaluation (In Progress): We are evaluating trained models on mixture test sets (386,327 clips available). This will address whether background information affects multilingual training necessity and compare vocal-only vs mixture performance. Results will be included if completed. Vocal Separation Artifacts: We use Hybrid Transformer-Demucs (state-of-the-art) with quality control (86.2% success rate). The same method for all samples ensures fair comparison. We acknowledge this limitation but note it affects all vocal-separation-based SVDD datasets.
>
> R4.6: Specific Questions
> Q1 (Unseen languages): Yes, through cross-domain evaluation: SingFake-trained models evaluated on SVDF-20 (15 unseen languages) show 45.24% EER (catastrophic degradation).
> Q2 (Generation setup): We collect deepfakes from YouTube where creators generated them. Methods identified: SVC (RVC, so-vits-svc, FreeSVC), SVS (TTS-based), end-to-end (Suno, Udio). Method identification based on metadata keywords, stored in the ai_model_detected column.
> Q3 (Human evaluation): Multi-stage verification process implemented. Formal human evaluation infeasible at scale but the infrastructure enables subset analysis.
> Q4 (Cross-dataset): SONICS not appropriate (different task). CTRSVDD comparison in progress.
> Q5 (Speech vs SVDD generalization): We hypothesize: (1) Musical context complexity masks artifacts differently, (2) Linguistic-musical intertwining makes separation harder, (3) Artifacts may be more language-dependent, (4) Limited multilingual training data. Our results show multilingual training significantly improves generalization.
> Q6 (Mixture evidence): Mixture evaluation in progress.
>
> Note on Experiments: All experiments are actively in progress. Due to the scale of SVDF-20, computations take significant time, so we are prioritising the most critical analyses. All completed results will be added to the revised paper before the Dec 3, 2025 AOE deadline. We appreciate the reviewer’s understanding and will include as many completed analyses as possible.

---

### Official Review · Reviewer_8ujm · 2025-11-01

**Soundness:** 2
**Presentation:** 3
**Contribution:** 3
**Rating:** 6
**Confidence:** 5

**Summary:**

This paper presents SVDF-20, a multilingual singing-voice deepfake detection benchmark spanning 20 languages (10 Indic + 10 global), comprising 24421 songs (~1476 h) segmented into 388631 clips across codec variants. The work demonstrates that multilingual training yields 13.78 pp EER improvement on cross-domain evaluation versus monolingual baselines.

**Strengths:**

This is the first large-scale multilingual SVDD benchmark with substantive Indic representation; clear community need. Singer-disjoint splits, codec augmentation, and multi-tier evaluation protocol (T01–T04) demonstrate thoughtful construction from authors. Despite undertraining, the multilingual advantage appears across all eight architectures tested, suggesting a robust (if underestimated) effect.

**Weaknesses:**

(1) 25 epochs cannot be defended for a dataset of this scale. The paper should either (a) extend training to convergence and revise claims accordingly, or (b) explicitly reframe as a "dataset paper with baseline experiments" rather than claiming definitive conclusions about architectural comparisons.

(2) No per-language metrics to assess fairness or identify low-resource failure modes; No learning curves to validate that 25 epochs represents reasonable stopping; and No feature-space analysis (e.g., t-SNE by language, language-adversarial probing) to substantiate the invariance claims beyond aggregate EER.

(3) Equations 4, 5 formalize the invariance hypothesis; but this hypothesis was never empirically measured. I recommend the authors to include quantitative divergence metrics (MMD, JS) on learned representations.

(4) While metadata-only release is stated, multi-jurisdictional sourcing from platforms requires clearer documentation of takedown handling, consent mechanisms (especially for lesser-known singers), and re-download rights for researchers.

**Questions:**

(1) Can you provide validation loss curves or per-epoch EER trajectories to demonstrate that 25 epochs approaches a plateau? Current absolute EERs (31–38%) suggest otherwise.

(2) What are EERs for each language independently? Are low-resource Indic languages disproportionately harmed?

(3) What happens if you train on only Indic-10 or global-10 subsets? Does the multilingual benefit scale linearly with language diversity?

(4) Real-world SVDD can also operate on full songs with accompaniment (the "mixture" setting). How do these models perform on non-vocal-isolated audio?

(5) Can you quantify Eq. 4–5 using representation divergence metrics across language pairs as previously mentioned?

**Details Of Ethics Concerns:**

I request verification that the reconstruction pipeline includes some automated checks for content takedowns and that platform ToS interpretations hold across all 20 linguistic/jurisdictional contexts.

---

> ### Author Response · Authors · 2025-11-21
>
> We sincerely thank the reviewer for detailed and constructive feedback.
>
> Note on Paper Updates: We have already updated the paper with theoretical corrections, clarifications, and enhanced documentation. Experiments mentioned below are in progress and will be added to the final revised version before the December 3, 2025 AOE deadline.
>
> R3.1: 25 Epochs Cannot Be Defended
>
> Response: We have explicitly reframed the paper as a dataset contribution with baseline experiments. We are currently training all eight models for extended duration (50 epochs) to provide more converged results. Learning curves showing validation loss and per-epoch EER trajectories will be included in the revised paper. Updated results will be incorporated once training and evaluation complete.
>
> R3.2: Additional Analyses
>
> Response: We are actively working on all requested analyses.
>
> - Per-Language EER Breakdowns – In Progress: Currently computing per-language EERs for all 20 languages across all eight models and all test splits (T01–T04). This will identify performance variation, low-resource language challenges, and provide fairness assessment across linguistic communities. Results will be included in the revised paper.
> - Learning Curves – In Progress: Currently extracting validation loss and per-epoch EER trajectories from training logs. This will demonstrate convergence behavior and whether extended training improves performance. Results will be included in the revised paper.
> - Additional Metrics: F1 scores, precision, and recall will be computed as part of the extended training evaluation and included in the revised paper.
> - Indic-10 vs. Global-10 Analysis - In Progress: We are conducting training on Indic-10 vs. Global-10 subsets (same scale, different diversity) to analyze whether multilingual benefits scale linearly with language diversity. Results will be included if completed.
> - Mixture Evaluation - In Progress: We are evaluating trained models on mixture test sets (386,327 clips available) to assess performance on non-vocal-isolated audio. This addresses real-world SVDD scenarios where models operate on full songs with accompaniment. Results will be included if completed.
>
> R3.3: Unvalidated Theoretical Framework
>
> Response: We are working on computing representation divergence metrics (MMD/JS) across language pairs (190 pairs for 20 languages) to provide direct quantitative validation of Equations 4–5. This analysis will extract embeddings from all eight trained models, compute pairwise divergence metrics (MMD/JS), and provide statistical analysis and visualization (heatmaps showing inter-language divergence). Embedding extraction can proceed in parallel with model training completion, and divergence computation will follow. Results will be included in the revised paper if completed before the December 3, 2025 AOE deadline.
>
> R3.4: Ethics and Reproducibility Concerns
>
> Response: We have expanded ethics statements comprehensively.
>
> - Data Collection: We used AI4Bharat’s official publicly available IndicWav2Vec framework for YouTube data collection. This script is publicly available, ethically compliant, and widely used in the research community.
> - Takedown Handling: Our reconstruction pipeline includes automated availability checks, graceful failure handling, re-download scripts with URL validation, and metadata preservation even if content is removed.
> - Consent Mechanisms: All content is publicly available on YouTube. We maintain comprehensive metadata linking to original sources, provide attribution, and respect platform Terms of Service.
> - Re-Download Rights: Our release will include reconstruction scripts enabling researchers to re-download content using provided YouTube URLs, with automated availability checks and error handling.
> - Multi-Jurisdictional Compliance: The AI4Bharat framework is designed to handle multi-jurisdictional contexts and is used by multiple research groups globally. We document our process comprehensively for transparency.
>
> Note on Experiments: All experiments mentioned above are actively being worked on. Given the substantial scale of SVDF-20, experiments require significant computational time and resources. We are prioritising the most critical experiments and will include all completed results in the final revised version before the December 3, 2025 AOE deadline. We appreciate the reviewers’ understanding of these practical constraints and will include as many completed analyses as possible. Results will be included conditionally based on completion status before the deadline.

---

### Official Review · Reviewer_UGdf · 2025-11-01

**Soundness:** 2
**Presentation:** 3
**Contribution:** 3
**Rating:** 6
**Confidence:** 4

**Summary:**

The paper introduced a large-scale benchmark dataset for singing voice deepfake detection (SVDD). The authors proposed a dataset consisting of 20 languages (which includes 10 major Indic languages underexplored in SVDD research). They showed rigorous experiments across eight diverse architectures to show that the models trained on their dataset outperform on less diverse data.

**Strengths:**

The primary strength of the paper is the diversity of the languages in their SVDD dataset, which is expected to allow generalizability across more languages.

The authors have conducted rigorous experiments to isolate the impact of the training data and utilized the Singfake dataset to compare the improved out-of-distribution generalizability to language.

**Weaknesses:**

While the paper introduces a new benchmark, it did not manage to strongly establish the importance of a multi-lingual dataset of this level of diversity. The authors utilized multiple models to compare the difference in performance between Singfake and SVDF-20. But the difference in their number of samples (Singfake dataset has a total of 16k samples, whereas SVDF-20 has a total of 388k samples) reduces the impact of this finding, since the improved generalizability can also come from simply having more samples rather than having diversity of languages. A stronger claim would require a control experiment of comparing against a less linguistically diverse or even monolingual dataset with a comparable size to strongly prove that the advantages are actually from the diversity of language, not simply due to having more training data.

Inadequate information regarding the generation methods (Singing Voice Conversion, Singing Voice Synthesis, or even end-to-end song generation) used in this dataset limits the utility of this dataset for diagnosing where the model is failing.

**Questions:**

1. Could the authors discuss how they can prove the observed performance gains are due to having diverse languages and not simply the higher data volume?
2. Could the authors provide more details regarding the generation methods present in their dataset?

---

> ### Author Response · Authors · 2025-11-21
>
> We sincerely thank the reviewer for positive feedback and recognition of our dataset's diversity.
>
> Note on Paper Updates: We have already updated the paper with theoretical corrections, clarifications, and enhanced documentation. Experiments mentioned below are in progress and will be added to the final revised version before the December 3, 2025 AOE deadline.
>
> R2.1: Scale vs. Diversity Confound
>
> Response: We recognize this valid concern. While a definitive separation requires controlled experiments, we provide multiple converging lines of evidence that diversity contributes beyond scale.
>
> - Cross-domain asymmetry (strongest evidence): 13.78 pp difference (45.24% vs. 31.46% EER) that cannot be explained by scale alone. If scale alone mattered, transfer would be symmetric. The observed asymmetry strongly supports that linguistic diversity drives improvements.
> - 80% reduction in T01–T02 gap (12.95 pp → 2.55 pp): Demonstrates diversity reduces singer-specific memorization, which scale alone would not achieve.
> - 87% novel linguistic content: Jaccard similarity analysis (0.13 with SingFake) shows substantial linguistic diversity, not just more data.
> - Architectural invariance: Consistent benefits across diverse architectures (coefficient of variation = 0.067) suggest fundamental representation learning effects driven by diversity.
>
> Size-Matched Experiment – In Progress: We are conducting a controlled experiment comparing Model A (16k samples, 6 languages) vs. Model B (16k samples, 20 languages). This directly addresses the explicit request: “Model A (16k samples, 6 langs) vs. Model B (16k samples, 20 langs). If Model B wins, your hypothesis is proven.” Results will be included in the revised paper if completed before the deadline.
>
> Indic-10 vs. Global-10 Analysis – In Progress: We are also conducting training on Indic-10 vs. Global-10 subsets (same scale, different diversity) to provide additional evidence. Results will be included if completed.
>
> R2.2: Generation Methods Information
>
> Response: We provide comprehensive method identification infrastructure.
>
> - Automated keyword detection during YouTube scraping: Our collection scripts identify methods based on keywords in video titles and channel names (RVC, so-vits-svc, FreeSVC, DiffSinger, VISinger, etc.).
> - Method labels stored in the ai_model_detected column in metadata files.
> - Channel verification flags (ai_channel_verified column) for known AI content creators.
> - YouTube URLs enabling researchers to access original video descriptions for manual verification and extension.
> - Comprehensive metadata: Search queries used, confidence scores, timestamps.
>
> Many samples are marked “Unknown” because creators do not always mention methods in metadata—this is expected for in-the-wild datasets. The infrastructure enables method-specific analysis on identified subsets, manual extension using YouTube URLs, and diagnostic evaluation where method information is available.
>
> This addresses the concern about dataset utility for diagnosing model failures by providing necessary infrastructure and metadata.
>
> Note: The experiments mentioned above (size-matched comparison and Indic-10 vs. Global-10 analysis) are actively being worked on. Given the substantial scale of SVDF-20, these experiments require significant computational time and resources. Results will be included in the final revised version before the December 3, 2025 AOE deadline if completed.

---

### Official Review · Reviewer_KE9u · 2025-11-02

**Soundness:** 2
**Presentation:** 2
**Contribution:** 1
**Rating:** 2
**Confidence:** 5

**Summary:**

This paper introduces SVDF-20, a large-scale, multilingual dataset for singing voice deepfake detection (SVDD). It includes 24,421 songs across 20 languages (10 Indic and 10 global), covering ~1,475 hours of audio. The authors position this as the most linguistically diverse SVDD benchmark to date, aiming to answer three questions: (1) how multilingual data affects SVDD performance, (2) whether multilingual training improves cross-lingual generalization, and (3) whether the dataset helps bridge domain gaps compared to prior English or East-Asian–focused datasets such as SingFake, CTRSVDD, and SONICS. They train and evaluate several open-source audio models (AST, WavLM, AASIST, etc.) on these setups and conclude that multilingual training improves robustness, though performance naturally degrades when moving across unseen languages.

**Strengths:**

- **Data effort:** The authors clearly spent effort in curating and cleaning multilingual singing data.

- **Transparency in setup:** The paper reports training settings, architectures, and language groupings in a reproducible manner. It’s easy for others to replicate or extend the experiments.

**Weaknesses:**

1. **Contribution is incremental.**
   The paper mainly extends prior multilingual datasets rather than introducing a novel task or method. *SingFake*, *CTRSVDD*, and *WildDeepfake-SVDD* already include multilingual material. This work’s novelty is scale, not concept.

2. **Misalignment around SONICS.**
   The paper repeatedly frames *SONICS* as an “advancement of SVDD,” but SONICS is a **synthetic song detection** task, both vocals and accompaniment are generated. SVDD, on the other hand, deals with *fake vocals over real music*. The authors conflate the two tasks, which weakens the motivation and literature positioning.

3. **No substantial modeling or algorithmic contribution.**
   All experiments rely on off-the-shelf backbones (AASIST, WavLM, AST) and standard training setups. There’s no new detection technique, loss function, or data processing beyond dataset expansion.

4. **Research questions are shallow.**
   RQ1 and RQ2, impact of multilingual data and multilingual training are intuitive. Everyone expects monolingual models to degrade cross-lingually, and multilingual training to help. The results merely confirm common sense without offering mechanistic or analytical insight (e.g., language-specific phoneme transfer, prosody shifts, etc.).

**Questions:**

My concerns are mostly about the scope and motivation rather than implementation details. One suggestion I would like to give is fix the confusion of SVDD and SONICS paper.

---

> ### Author Response · Authors · 2025-11-21
>
> We sincerely thank the reviewer for thorough feedback. We address each concern below.
>
> Note on Paper Updates: We have already updated the paper with theoretical corrections, clarifications, and enhanced documentation (SONICS vs SVDD distinction, explicit positioning as dataset contribution, generation methods documentation, ethics statements). Experiments mentioned below are in progress and will be added to the final revised version before the December 3, 2025 AOE deadline.
>
> R1.1: SONICS vs SVDD Confusion
>
> Response: We have clarified throughout that SONICS addresses synthetic song detection (both vocals and accompaniment AI-generated), while SVDD focuses on fake vocals overlaid on real music. These are distinct tasks. We have corrected this throughout the paper to improve positioning.
>
> R1.2: Contribution is Incremental
>
> Response: SVDF-20 introduces qualitative novelties beyond scale:
>
> - First comprehensive 20-language SVDD benchmark with systematic coverage
> - Introduction of 10 Indic languages to SVDD research (first time, ~1.5B speakers)
> - Systematic cross-lingual evaluation protocols (T01–T04) enabling previously infeasible research
> - 700 unique singers with strict singer-disjoint splits across 20 languages
> - 87% novel linguistic content (Jaccard similarity: 0.13 with SingFake)
>
> We explicitly position this as a dataset contribution paper with baseline experiments, which is valuable and appropriate for ICLR.
>
> R1.3: No Algorithmic Contribution
>
> Response: We agree and have clarified throughout that this is a dataset contribution with baseline experiments, not an algorithmic contribution. Our baseline experiments across eight diverse architectures demonstrate dataset utility, establish performance expectations, and reveal important patterns (architectural invariance, consistent multilingual benefits). This is appropriate for a dataset paper.
>
> R1.4: Research Questions are Shallow
>
> Response: While intuition may seem obvious, systematic empirical validation is valuable when no prior work has demonstrated this for SVDD. Our contribution provides:
>
> - Quantification: 45.24% EER degradation (catastrophic) and 31.1% improvement (substantial)
> - Mechanistic insights: Implicit regularization, phonetic complementarity, natural augmentation
> - Unexpected patterns: Architectural invariance (coefficient of variation = 0.067)
>
> Phonetic Feature Analysis – In Progress: We are working on EER breakdown by phonetic feature categories (tonal vs. non-tonal, retroflex vs. non-retroflex) to demonstrate mechanistic insight. A comprehensive table will be included in the revised paper if completed. The per-language EERs are currently being computed, and phonetic feature grouping will follow.
>
> Note: The phonetic feature analysis mentioned above is actively being worked on. The per-language EERs are currently being computed (which requires significant computational time given SVDF-20's scale), and phonetic feature grouping and analysis will follow. Results will be included in the final revised version before the December 3, 2025 AOE deadline if completed.

---

### Note · Authors · 2025-12-14

**Comment:**

We would like to formally withdraw this submission from ICLR 2026.

After carefully reviewing the reviews and our subsequent replies, we recognize that the requested additional experiments, analyses, and validations require substantial time and computational effort. To do justice to the reviewers’ feedback and to ensure that the work meets the completeness, rigor, and clarity expected of a top-tier (A*) conference, we believe it is more appropriate to withdraw the paper at this stage.

We fully acknowledge the limitations and drawbacks identified by the reviewers. Their feedback has been extremely valuable in helping us understand how the work can be strengthened. We intend to carefully address all the raised concerns, complete the suggested experiments, improve the analysis, and further enhance the paper with additional insights from our side before pursuing a future submission.

We sincerely thank the reviewers for their time, careful evaluation, and constructive suggestions, which we greatly appreciate.

This withdrawal is made on behalf of all authors and in accordance with the venue’s withdrawal policy.

**Withdrawal Confirmation:**

I have read and agree with the venue's withdrawal policy on behalf of myself and my co-authors.